# SCALING FOR TRAINING-TIME AND POST-HOC OUT-OF-DISTRIBUTION DETECTION ENHANCEMENT

**Kai Xu**[1]**, Rongyu Chen**[1]**, Gianni Franchi**[2]**, Angela Yao**[1]
[1]National University of Singapore
[2]U2IS, ENSTA Paris, Institut polytechnique de Paris

{kxu,rchen,ayao}@comp.nus.edu.sg   gianni.franchi@ensta-paris.fr

## ABSTRACT

Activation shaping has proven highly effective for identifying out-of-distribution (OOD) samples post-hoc. Activation shaping prunes and scales network activations before estimating the OOD energy score; such an extremely simple approach achieves state-of-the-art OOD detection with minimal in-distribution (ID) accuracy drops. This paper analyzes the working mechanism behind activation shaping. We directly show that the benefits for OOD detection derive only from scaling, while pruning is detrimental. Based on our analysis, we propose SCALE, an even simpler yet more effective post-hoc network enhancement method for OOD detection. SCALE attains state-of-the-art OOD detection performance without *any* compromises on ID accuracy. Furthermore, we integrate scaling concepts into learning and propose **I**ntermediate Tensor **SH**aping (ISH) for training-time OOD detection enhancement. ISH achieves significant AUROC improvements for both near- and far-OOD, highlighting the importance of activation distributions in emphasizing ID data characteristics. Our code and models are available at https://github.com/kai422/SCALE.

## 1 INTRODUCTION

Out-of-distribution (OOD) detection for neural networks distinguishes samples which deviate from the training distribution. Standard OOD detection concerns semantic shifts (Yang et al., 2022; Zhang et al., 2023), where OOD data is defined as test samples from semantic categories unseen during training. Ideally, a neural network should be able to reject such samples as being OOD, while still maintaining strong performance on in-distribution (ID) test samples belonging to seen training categories.

Methods for detecting OOD samples work by scoring network outputs such as logits or softmax values (Hendrycks & Gimpel, 2017; Hendrycks et al., 2022), by making post-hoc network adjustments during inference (Sun & Li, 2022; Sun et al., 2021; Djurisic et al., 2023), or by adjusting model training (Wei et al., 2022; Ming et al., 2023; DeVries & Taylor, 2018). These approaches can be used either independently or in conjunction with one another. Typically, post-hoc adjustment together with OOD scoring is the preferred combination since it can discern OOD samples with minimal ID drop and can be applied directly to already-trained models off the shelf. Examples of post-hoc adjustment methods include ReAct (Sun et al., 2021), DICE (Sun & Li, 2022) and more recently, ASH (Djurisic et al., 2023).

On the surface, post-hoc methods take different and sometimes even contradictory approaches. For example, ReAct rectifies penultimate activations which exceed a threshold. ASH, on the other hand, prunes penultimate activations that are too low while amplifying remaining activations. ASH currently achieves state-of-the-art performance but lacks a comprehensive explanation of its underlying operational principles.

This work seeks to understand the working principles behind ASH. Through experimental observations and mathematical derivations, we demonstrate that ID and OOD datasets respond differently to pruning, attributable to their unique activation distributions. We also demonstrate the significant role of activation scaling in enhancing OOD detection while highlighting that low activation pruning

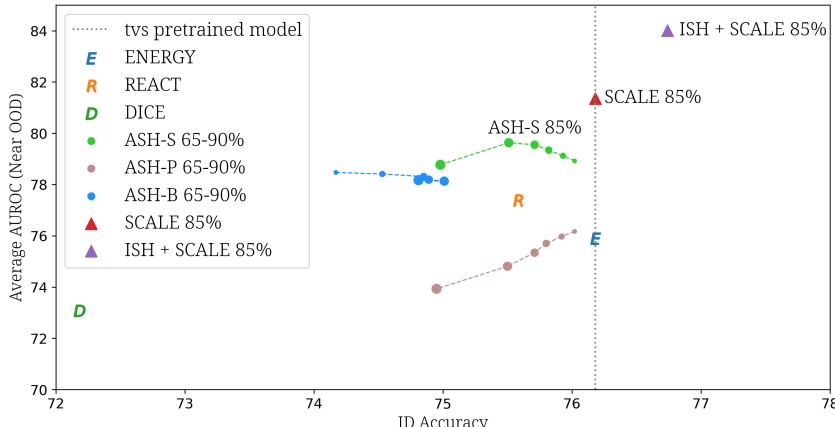

Figure 1: **ID-OOD Trade-off on ImageNet on Near-OOD Dataset.** Unlike existing methods such as ASH, ReAct, and DICE, our proposed SCALE does not have any ID accuracy trade-off while improving OOD detection accuracy. Our training method, ISH, achieves outstanding OOD results by emphasizing the training of samples with high ID characteristics.

hinders OOD detection. This understanding leads directly to our proposed approach, SCALE, for post-hoc OOD detection enhancement; notably, SCALE attains state-of-the-art results, achieving significant enhancements without compromising ID accuracy.

Our study on activation distributions highlights the importance of activation magnitude as a metric to assess a sample's ID nature. We further propose to incorporate this finding by scaling the activation to emphasize the optimization priors on the sample's ID-ness during network training. To that end, we propose intermediate tensor shaping (ISH), which explores the concept of ID-ness as a learning rate weighting factor at the penultimate layer. Remarkably, ISH achieves outstanding performance in both near-OOD and far-OOD detection tasks, with only one-third of the training effort required compared to current state-of-the-art approaches.

Our contributions can be summarized as follows:

- We analyze and explain the working principles of pruning and scaling for OOD detection and reveal that the benefits come only from scaling, while pruning, in some scenarios, may actually hurt OOD detection.

- Based on our analysis, we devise SCALE, a new post-hoc network enhancement method for OOD detection, which achieves state-of-the-art results on OOD detection without any ID accuracy trade-off.

- We incorporate scaling concepts into learning to emphasize the training sample's ID characteristics by introducing ISH. ISH is a lightweight and innovative method that uses ID-ness as a learning rate weighting factor at the penultimate layer; it surpasses state-of-the-art with a large margin on both near- and far-OOD.

## 2  RELATED WORKS

**OOD scoring methods** indicate how likely a sample comes from the training distribution, *i.e.* is in-distribution, based on the sample's features or model outputs. From a feature perspective, Lee et al. (2018) proposed to score a sample via the minimum Mahalanobis distance of that sample's features to the nearest ID class centroid. For model outputs, two common variants are based on the maximum softmax prediction (Hendrycks & Gimpel, 2017) and the maximum logit scores (Hendrycks et al., 2022). The raw softmax or logit scores are  susceptible to overconfidence so  Liu et al. (2020) proposed to use an energy-based function to transform the logits as an improved score. A key benefit of deriving OOD scores from feature or model outputs is that it does not impact the model or the inference procedure, so the ID accuracy will not be affected.

**Post-hoc model enhancement methods** modify the inference procedure to improve OOD detection and are often used together with OOD scoring methods. Examples include ReAct (Sun et al., 2021), which rectifies the penultimate activations for inference, DICE (Sun & Li, 2022), which sparsifies the network's weights in the last layer, and ASH (Djurisic et al., 2023), which scales and prunes the penultimate activations. Each method is combined with energy-based scoring (Liu et al., 2020) to detect the OOD data. While effective at identifying OOD data, these methods may have a reduced ID accuracy as the inference procedure is altered. Our proposed SCALE is also a post-hoc model enhancement; however, as we shape each sample's activations by a constant value, the argmax remains unaffected and preserves the ID accuracy.

**Training-time model enhancement methods** aim to make OOD data more distinguishable directly at training. Strategies include the incorporation of additional network branches (DeVries & Taylor, 2018), alternative forms of training (Wei et al., 2022), or data augmentation (Pinto et al., 2022; Hendrycks et al., 2020). The underlying assumption behind these techniques is that training can provide more discriminative features for OOD detection. A significant drawback of training-time enhancement is the additional computational cost. For example, AugMix (Hendrycks et al., 2020) requires double training time and extra GPU memory cost. Our intermediate tensor shaping (ISH) improves the OOD detection with one-third of the computational cost compared to the most lightweight method, without modifying the model architecture.

**Activation and intermediate tensor shaping** have been explored in deep learning for various purposes. Perhaps the best-known example is Dropout (Srivastava et al., 2014), which sparsifies activations for regularization. Similar ideas have been applied by Li et al. (2023) for transformer regularization, as well as for efficient training and inference. (Kurtz et al., 2020; Chen et al., 2023b). Shaping operations on intermediate tensors differ from those applied to activations. Activation shaping affects both forward-pass inference and backward-gradient computation during training. In contrast, shaping intermediate tensors exclusively influences the backward gradient computation. Since intermediate tensors tend to consume a significant portion of GPU memory, techniques for shaping intermediate tensors have gained widespread use in memory-efficient training, while leaving the forward pass unaltered (Evans & Aamodt, 2021; Liu et al., 2022; Chen et al., 2023a).

## 3 ACTIVATION SCALING FOR POST-HOC MODEL ENHANCEMENT

We start by presenting the preliminaries of Out-of-Distribution (OOD) detection in Sec. 3.1, to set the stage for our subsequent analysis of the ASH method in Sec. 3.2. The results of our analysis directly leads to our own post-hoc OOD detection enhancement in Sec. 3.3. Finally, we introduce our intermediate tensor shaping approach for training-time OOD detection enhancement in Sec. 3.4.

### 3.1 PRELIMINARIES

While OOD is relevant for many domains, we follow previous works (Yang et al., 2022) and focus specifically on semantic shifts in image classification. During training, the classification model is trained only with ID data $\mathcal{D}_{\text{ID}}$ that fall into a pre-defined set of $K$ classes $\mathcal{Y}_{\text{ID}}$: $\forall (\boldsymbol{x}, y) \sim \mathcal{D}_{\text{ID}}, y \in \mathcal{Y}_{\text{ID}}$. During inference, in addition to ID samples, there are also samples from OOD data $\mathcal{D}_{\text{OOD}}$; the latter are samples from classes unobserved during training, *i.e.* $\forall (\boldsymbol{x}, y) \sim \mathcal{D}_{\text{OOD}}, y \notin \mathcal{Y}_{\text{ID}}$.

Consider a neural network consisting of two parts: a feature extractor $f(\cdot)$, and a linear classifier parameterized by weight matrix $\mathbf{W} \in \mathbb{R}^{K \times D}$ and a bias vector $\boldsymbol{b} \in \mathbb{R}^K$. The network logit $\boldsymbol{z}$ can be mathematically represented as:

$$\boldsymbol{z} = \mathbf{W} \cdot \boldsymbol{a} + \boldsymbol{b}, \qquad \boldsymbol{a} = f(\boldsymbol{x}), \tag{1}$$

where $\boldsymbol{a} \in \mathbb{R}^D$ is the $D$-dimensional feature vector in the penultimate layer of the network and $\boldsymbol{z} \in \mathbb{R}^K$ is the logit vector from which the class label can be estimated by $\hat{y} = \arg\max(\boldsymbol{z})$. In line with other OOD literature (Sun et al., 2021), an individual dimension of feature $\boldsymbol{a}$, denoted with index $j$ as $\boldsymbol{a}_j$, is referred to as an "activation".

For a given test sample $\boldsymbol{x}$, an OOD score can be calculated to indicate the confidence that $\boldsymbol{x}$ is in-distribution. By convention, scores above a threshold $\tau$ are ID, while those equal or below are considered OOD. A common setting is the energy-based OOD score $S_{EBO}(\boldsymbol{x})$ together with indicator function $G(\cdot)$ that applies the thresholding (Liu et al., 2020):

$$G(\boldsymbol{x}; \tau) = \begin{cases} 0 & \text{if } S_{EBO}(\boldsymbol{x}) \leq \tau \quad (OOD), \\ 1 & \text{if } S_{EBO}(\boldsymbol{x}) > \tau \quad (ID), \end{cases} \qquad S_{EBO}(\boldsymbol{x}) = T \cdot \log \sum_{k}^{K} e^{\boldsymbol{z}_k/T}, \qquad (2)$$

where $T$ is a temperature parameter and $k$ is the logit index for the $K$ classes.

## 3.2 Analysis on ASH

One state-of-the-art method for OOD detection is ASH (Djurisic et al., 2023). ASH stands for activation shaping and applies a rectified scaling to the feature vector $\boldsymbol{a}$ post-hoc during inference. Activations in $\boldsymbol{a}$ up to the $p^{\text{th}}$ percentile across the $D$ dimensions are rectified ("pruned" in the original text); activations above the $p^{\text{th}}$ percentile are scaled. More formally, ASH introduces a shaping function $s_f$ that is applied to each activation $\boldsymbol{a}_j$ for a given sample. If we define $P_p(\boldsymbol{a})$ as the value of the $p^{\text{th}}$ percentile of the elements in feature $\boldsymbol{a}$, ASH produces the "enhanced" logit $\boldsymbol{z}_{\text{ASH}}$:

$$\boldsymbol{z}_{\text{ASH}} = \mathbf{W} \cdot (\boldsymbol{a} \circ s_f(\boldsymbol{a})) + \boldsymbol{b}, \quad \text{where } s_f(\boldsymbol{a})_j = \begin{cases} 0 & \text{if } \boldsymbol{a}_j \leq P_p(\boldsymbol{a}), \\ \exp(r(\boldsymbol{a})) & \text{if } \boldsymbol{a}_j > P_p(\boldsymbol{a}). \end{cases} \qquad (3)$$

In the equation above, $\circ$ denotes an element-wise multiplication. The scaling factor $r$ is defined as the ratio of the sum of all activations versus the sum of un-pruned activations in $\boldsymbol{a}$:

$$r(\boldsymbol{a}) = \frac{Q(\boldsymbol{a})}{Q_p(\boldsymbol{a})}, \qquad \text{where } Q(\boldsymbol{a}) = \sum_{j}^{D} \boldsymbol{a}_j \qquad \text{and } Q_p(\boldsymbol{a}) = \sum_{\boldsymbol{a}_j > P_p(\boldsymbol{a})} \boldsymbol{a}_j. \qquad (4)$$

Since $Q_p(\boldsymbol{a}) \leq Q(\boldsymbol{a})$, the factor $r(\boldsymbol{a}) \geq 1$; the higher the percentile $p$, *i.e.* the greater the extent of pruning, the smaller $Q_p(\boldsymbol{a})$ w.r.t. $Q(\boldsymbol{a})$ and the larger the scaling factor $r(\boldsymbol{a})$. To distinguish OOD data, ASH passes the logit from Eq. 3 to the score and indicator function as given in Eq. 2.

ASH is highly effective but the original paper has no explanation of the working mechanism[1]. We analyze the rectification and scaling components of ASH below and reveal that scaling helps to separate ID versus OOD energy scores, while rectification or pruning has an adverse effect.

| Dataset | $p$ value |
|---|---|
| ImageNet | 0.296 |
| SSB-hard | 0.262 |
| NINCO | 0.181 |
| iNaturalist | 0.083 |
| Textures | 0.099 |
| OpenImage-O | 0.155 |

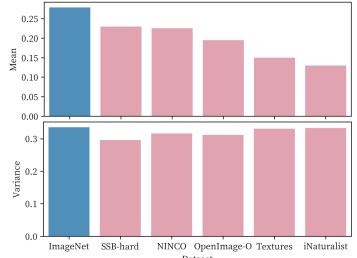

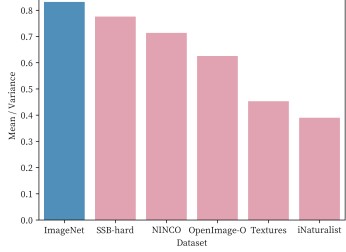

Table 1: Average $p$-values for all samples under the Chi-square test; values greater than 0.05 verify that a Gaussian assumption is reasonable.

Figure 2: Mean and variance of pre-ReLU activations for ID (blue) vs. OOD datasets (pink).

Figure 3: $\mu/\sigma$ of pre-ReLU activations for ID (blue) vs. OOD (pink).

**Assumptions:** Our analysis is based on two assumptions. First, we assume that the penultimate activations of ID and OOD samples follow rectified Gaussian distributions. The Gaussian assumption is commonly used in the literature (Sun et al., 2021) and we verify it empirically in Tab. 1. The rectification of the Gaussian follows naturally if a ReLU is applied as the final operation of the penultimate layer. Secondly, we assume that the mean of ID activations is higher than that of OOD activations, with the variances remaining equivalent; this assumption is supported by Liu et al. (2020),

---

[1]In fact, the authors put forth a call for explanation in their Appendix L.

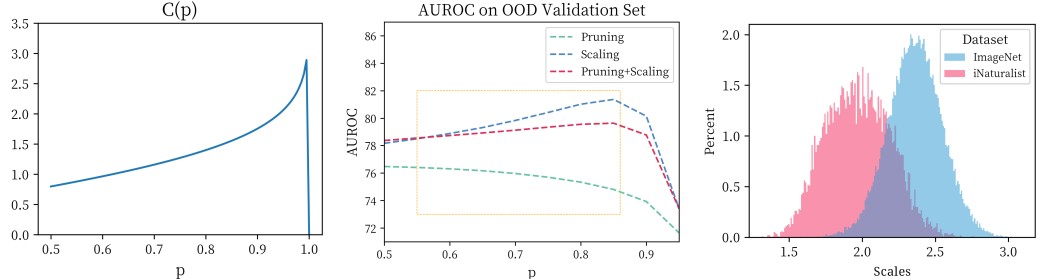

Figure 4: (a) The relationship between the parameter $C(p)$ and the percentile $p$. A higher value of $C(p)$ indicates better separation of scales between ID and OOD datasets. (b) AUROC vs. percentile $p$. Up to $p = 0.85$, as highlighted by the orange box, AUROC for scaling increases while for pruning it decreases. The results of ASH sit between the two as the method is a combination of pruning plus scaling. (c) Histograms of scaling factor $r(\boldsymbol{a}) = Q(\boldsymbol{a})/Q_p(\boldsymbol{a})$ for the ID dataset (ImageNet) and OOD dataset (iNaturalist); the scales exhibit a clear separation from each other.

who suggested that well-trained networks have higher responses to samples resembling those seen in training. Combining these two assumptions, we can specify, for the ID and OOD rectified Gaussians parameterized by $(\mu^{ID}, \sigma^{ID})$ and $(\mu^{OOD}, \sigma^{OOD})$ respectively, $\frac{\mu^{ID}}{\sigma^{ID}} > \frac{\mu^{OOD}}{\sigma^{OOD}}$. Figure 2 and 3 visualize statistical corroboration of these assumptions; for the ID dataset ImageNet (blue bar), various OOD datasets (pink bars) all have a consistently smaller $\frac{\mu}{\sigma}$. Based on these assumptions, we make the following proposition:

**Proposition 3.1.** *Assume that ID activations $\boldsymbol{a}_j^{ID} \sim \mathcal{N}^R(\mu^{ID}, \sigma^{ID})$ and OOD activations $\boldsymbol{a}_j^{OOD} \sim \mathcal{N}^R(\mu^{OOD}, \sigma^{OOD})$ where $\mathcal{N}^R$ denotes a rectified Gaussian distribution. If $\mu^{ID}/\sigma^{ID} > \mu^{OOD}/\sigma^{OOD}$. There exists a range of percentiles $p$ for which a factor $C(p) = \frac{\varphi(\sqrt{2}\,\mathrm{erf}^{-1}(2p-1))}{1-\Phi(\sqrt{2}\,\mathrm{erf}^{-1}(2p-1))}$ is large enough such that $Q_p(\boldsymbol{a}^{ID})/Q(\boldsymbol{a}^{ID}) < Q_p(\boldsymbol{a}^{OOD})/Q(\boldsymbol{a}^{OOD})$.*

The proof of the proposition is given in Appendix A. Above, $\varphi$ and $\Phi$ denote the probability density function and cumulative distribution function of the standard normal distribution, respectively. The factor $C(p)$, plotted in Fig. 4a, relates the percentile of activations that distinguishes ID from OOD data. Specifically, the proposition states that the sum of $p^{\text{th}}$ percentile strongest activations for OOD data is higher (relatively) than for ID data.

**Rectification (Pruning)** sets activations smaller than $P_p(\boldsymbol{a})$ to 0. The relative reduction of activations can be expressed as:

$$D^{Pruning}(\boldsymbol{a}) = (Q(\boldsymbol{a}) - Q_p(\boldsymbol{a}))/Q(\boldsymbol{a}) = 1 - Q_p(\boldsymbol{a})/Q(\boldsymbol{a}). \tag{5}$$

Note that a reduction in activations also leads to a reduction in the energy score. Since $Q_p(\boldsymbol{a}^{ID})/Q(\boldsymbol{a}^{ID}) < Q_p(\boldsymbol{a}^{OOD})/Q(\boldsymbol{a}^{OOD})$, it directly implies that $D^{Pruning}(\boldsymbol{a}^{ID}) > D^{Pruning}(\boldsymbol{a}^{OOD})$, i.e. the decrease in ID samples will be greater than that in OOD samples. Building upon Remark 2 presented by ReAct (Sun et al., 2021), which demonstrates the proportionality between changes in logits and activations, it can be deduced that, for ID samples, the relative decrease in the final energy scores due to rectification will surpass that of OOD samples.

The result above shows that rectification or pruning creates a greater overlap in energy scores between ID and OOD samples, making them more difficult to distinguish. Empirically, this result is shown in Fig. 4b, where AUROC steadily decreases with stand-alone pruning as the percentile $p$ increases.

**Scaling** on the other hand behaves in a manner opposite to the derivation above. Again, given that $Q_p(\boldsymbol{a}^{ID})/Q(\boldsymbol{a}^{ID}) < Q_p(\boldsymbol{a}^{OOD})/Q(\boldsymbol{a}^{OOD})$, we have $r(\boldsymbol{a}^{ID}) > r(\boldsymbol{a}^{OOD})$. Figure 4c depicts the histograms for $r(\boldsymbol{a}^{ID})$ versus $r(\boldsymbol{a}^{OOD})$; they are well separated and therefore scale activations of ID and OOD samples differently. The relative increase in activation can be expressed as:

$$I^{Scaling}(\boldsymbol{a}) = r(\boldsymbol{a}) - 1 = Q(\boldsymbol{a})/Q_p(\boldsymbol{a}) - 1, \tag{6}$$

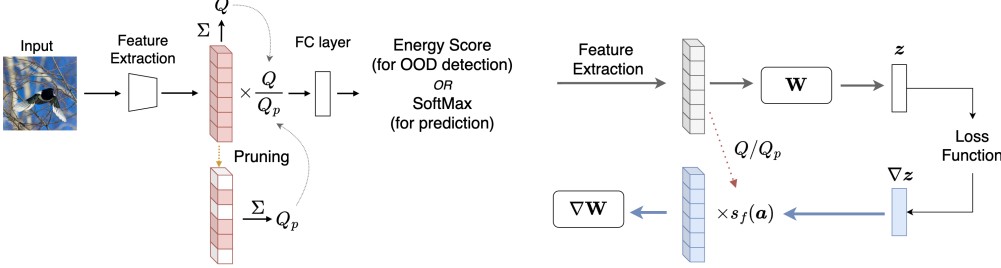

(a) Demonstration of SCALE post-hoc model improvement. We prune activations to calculate the scaling factor. The original activations are then multiplied by the computed scales before being fed into the fully connected layer.

(b) The process of ISH training. During training, we keep the forward pass unchanged. In the backward pass, we scale activations for parameter optimization weighted by $s_f(\boldsymbol{a}^i)$, which varies for different samples and reflects the sample's ID-ness.

Figure 5: Illustrations of our post-hoc model enhancement method SCALE and training-time model enhancement method ISH.

where we can get $I^{scaling}(\boldsymbol{a}^{ID}) > I^{scaling}(\boldsymbol{a}^{OOD})$. This increase is then transferred to logit spaces $\boldsymbol{z}$ and energy-based scores, thereby increasing the gap between ID and OOD samples.

**Discussion on percentile $p$:** Note that $C(p)$ does not monotonically increase with respect to $p$ (see Fig. 4a). When $p \approx 0.95$, there is an inflection point and $C(p)$ rapidly decreases. A similar inflection follows on the AUROC for scaling (see Fig. 4b), though it is not exactly aligned to $C(p)$ and drops off earlier. The difference is likely due to the approximations made to estimate $C(p)$. Furthermore, as $p$ gets progressively larger, fewer activations ($D = 2048$ total activations) are considered for estimating $r$, leading to unreliable logits for the energy score. Curiously, pruning also drops off, which we believe comes similarly from the extreme reduction in activations.

### 3.3 SCALE CRITERION FOR OOD DETECTION

From our analyses and findings above, we propose *SCALE*, a new post-hoc model enhancement method. As the name suggests, it shapes the activation with (only) a scaling:

$$\boldsymbol{z}' = \mathbf{W} \cdot (\boldsymbol{a} \circ s_f(\boldsymbol{a})) + \boldsymbol{b}, \quad \text{where } s_f(\boldsymbol{a})_j = \exp(r), \tag{7}$$

where $r$ is the same scaling factor as defined in Eq. 4 for ASH based on percentile $p$. Figure 5a provides an illustration of the SCALE procedure. Note that instead of pruning, SCALE retains and scales *all* the activations, *i.e.* $s_f(\boldsymbol{a})_j$ is never set to 0. Doing so has two benefits. First, it enhances the energy-score separation between ID and OOD samples. Secondly, scaling all the activations equally preserves the ordinality of the logits $\boldsymbol{z}'$ compared to $\boldsymbol{z}$. As such, the $\arg\max$ is not affected, and there is no trade-off for ID accuracy. Such is not the case with rectification, be it pruning, like in ASH or clipping, or like ReAct (see Fig. 1). Results in Tab. 2 and 3 verify that SCALE outperforms ASH-S (the variant with the best performance) on all datasets and model architectures.

### 3.4 INCORPORATING SCALE INTO TRAINING

In practice, the semantic shift from ID to OOD data may be ambiguous. For example, the iNaturalist dataset features different species of plants yet similar objects may be found in ImageNet. Our hypothesis is that, during training, we can further emphasize the impact of samples possessing more distinct in-distribution characteristics. A direct way is simply to reweight the loss of each training sample to be proportional to the "ID-ness" of each sample. To measure "ID-ness", we directly rely on the scale factor $r$, as defined in Eq. 4. To have a reliable measure on $r$, the network must already be well-trained; we therefore apply the re-weighting only for additional fine-tuning. To prevent the network from over-fitting to the upweighted samples, we apply the reweighting only for updates on the linear classifier weights $\mathbf{W}$ in the final layer, while other network weights are updated with a gradient based on the standard cross-entropy loss. Implementation-wise, it is more direct to simply scale the gradient for $\mathbf{W}$ as follows:

$$\mathbf{W}^{t+1} = \mathbf{W}^t - \eta \sum_i [(\boldsymbol{a}^i \circ s_f(\boldsymbol{a}^i))^\top \nabla \boldsymbol{z}^i], \qquad (8)$$

where $\nabla$ denotes the gradient regarding the cross entropy loss, $t$ denotes the training step $t$, and $\eta$ represents the learning rate. The activations $\boldsymbol{a}^i$ are already stored from the forward pass for estimating gradients so computing $r(\boldsymbol{a}^i)$ requires no additional memory and only minimal computational resources. As $\boldsymbol{a}^i$ are often referred to as intermediate tensors in the context of backpropagation, we name our method Intermediate tensor SHaping (ISH). Fig. 5b illustrates the gradient scaling procedure.

The methods described in Sec. 3.3 and Sec. 3.4 can be integrated, where ISH adjusts $f(\cdot)$ and $\mathbf{W}$ during the training phase, and SCALE modifies the activation $\boldsymbol{a}$ during inference, leading to improved results. We present experimental results in Sec. 4.3.

## 4 EXPERIMENTS

### 4.1 SETTINGS

We experiment with CIFAR10, CIFAR100 (Krizhevsky, 2009), and ImageNet-1K (Deng et al., 2009) ID data sources.

**CIFAR.** We used SVHN (Netzer et al., 2011), iSUN (Xu et al., 2015), Places365 (Zhou et al., 2018), and Textures (Cimpoi et al., 2014) as OOD datasets. For consistency with previous work, we use the same model architecture and pretrained weights, namely, DenseNet-101 (Huang et al., 2017), in accordance with the other post-hoc approaches DICE, ReAct, and ASH. Table 3 compares the FPR@95 and AUROC averaged across all four datasets; detailed results are provided in Appendix B.

**ImageNet.** We follow the OpenOOD v1.5 (Zhang et al., 2023) benchmark, which distinguishes between near- and far-OOD data. We employed SSB-hard (Vaze et al., 2022) and NINCO (Bitterwolf et al., 2023) as near-OOD datasets and iNaturalist (Horn et al., 2018), Textures (Cimpoi et al., 2014), and OpenImage-O (Wang et al., 2022) as far-OOD datasets. Our reported metrics are the average FPR@95 and AUROC values across these categories; detailed results are given in Appendix B. The OpenOOD benchmark includes improved hyperparameter selection with a dedicated OOD validation set to prevent overfitting to the testing set. Additionally, we provide results following the same dataset and test/validation split settings as ASH and ReAct in the appendix. We adopted the ResNet-50 (He et al., 2016) model architecture and obtained the pretrained network from the torchvision library.

**Metrics.** We evaluate with (1) FPR@95, which measures the false positive rate at a fixed true positive rate of 95%, where lower scores are better and (2) AUROC (Area under the ROC curve). It represents the probability that a positive ID sample will have a higher detection score than a negative OOD sample; higher scores indicate superior discrimination.

### 4.2 SCALE FOR POST-HOC OOD DETECTION

Comparison of ODD score methods and post-hoc model enhancement methods (separated with a solid line) on the ImageNet and CIFAR are illustrated in the Tab. 2 and 3. Notably, SCALE attains the highest OOD detection scores.

**OOD detection accuracy.** Compared to the current state-of-the-art ASH-S, SCALE demonstrates significant improvements on ImageNet – 1.73 on Near-OOD when considering AUROC. For FPR@95, it outperforms ASH-S by 2.27 and 0.33. On CIFAR10 and CIFAR100, SCALE has even greater improvements of 3.28 and 2.03 for FPR@95, as well as 0.91 and 0.68 for AUROC , respectively.

**ID accuracy.** One of SCALE's key advantages is it only applies linear transformations on features, so ID accuracy is guaranteed to stay the same. This differentiates it from other post-hoc enhancement methods that rectify or prune activations, which invariably compromise the ID accuracy. SCALE's performance surpasses ASH-S by a substantial margin of 0.67 on the ID dataset, ImageNet-1K. This capability is pivotal for establishing a unified pipeline that excels for ID and OOD.

**Comparison with TempScale.** Temperature scaling (TempScale) is widely used  for confidence calibration (Guo et al., 2017). SCALE and TempScale both leverage scaling for OOD detection,

| Model | Postprocessor | Near-OOD | | Far-OOD | | ID ACC |
|---|---|---|---|---|---|---|
| | | FPR@95 ↓ | AUROC ↑ | FPR@95 ↓ | AUROC ↑ | ↑ |
| ResNet-50 | MSP (Hendrycks & Gimpel, 2017) | 65.67 | 76.02 | 51.47 | 85.23 | 76.18 |
| | EBO (Liu et al., 2020) | 68.56 | 75.89 | 38.40 | 89.47 | 76.18 |
| | RMDS (Ren et al., 2021) | 65.04 | 76.99 | 40.91 | 86.38 | 76.18 |
| | MLS (Hendrycks et al., 2022) | 67.82 | 76.46 | 38.20 | 89.58 | 76.18 |
| | GEN (Liu et al., 2023) | 65.30 | 76.85 | 35.62 | 89.77 | 76.18 |
| | TempScale (Guo et al., 2017) | 64.51 | 77.14 | 46.67 | 87.56 | 76.18 |
| | ReAct (Sun et al., 2021) | 66.75 | 77.38 | 26.31 | 93.67 | 75.58 |
| | ASH-S (Djurisic et al., 2023) | 62.03 | 79.63 | 16.86 | 96.47 | 75.51 |
| | **SCALE (Ours)** | **59.76** | **81.36** | **16.53** | **96.53** | **76.18** |

Table 2: **OOD detection results on ImageNet-1K benchmarks.** Model choice and protocol are the same as existing works. SCALE outperforms other OOD score methods and post-hoc model enhancement methods, achieving the highest OOD detection scores and excelling in the ID-OOD trade-off. Detailed results for each dataset are given in Appendix B.

| Model | Postprocessor | CIFAR-10 | | CIFAR-100 | |
|---|---|---|---|---|---|
| | | FPR@95 ↓ | AUROC ↑ | FPR@95 ↓ | AUROC ↑ |
| DenseNet-101 | MSP (Hendrycks & Gimpel, 2017) | 54.18 | 91.18 | 83.76 | 72.84 |
| | EBO (Liu et al., 2020) | 36.55 | 92.53 | 81.34 | 77.39 |
| | ReAct (Sun et al., 2021) | 35.31 | 93.77 | 77.38 | 80.24 |
| | DICE (Sun & Li, 2022) | 30.21 | 93.09 | 62.00 | 83.15 |
| | ASH-S (Djurisic et al., 2023) | 21.12 | 95.25 | 47.89 | 87.76 |
| | **SCALE (Ours)** | **17.84** | **96.16** | **45.86** | **88.44** |

Table 3: **OOD detection results on CIFAR benchmarks.** SCALE outperforms all postprocessors. Detailed results for each dataset are in the appendix.

but with two distinctions. Firstly, TempScale directly scales logits for calibration, whereas SCALE applies scaling at the penultimate layer. Secondly, TempScale employs a uniform scaling factor for all samples, whereas SCALE applies a sample-specific scaling factor based on the sample's activation statistics. The sample-specific scaling is a crucial differentiator that enables the discrimination between ID and OOD samples. Notably, our SCALE model significantly outperforms TempScale in both Near-OOD and Far-OOD scenarios.

**SCALE with different percentiles $p$.** Table 2 uses $p = 0.85$ for SCALE and ASH-S, which is verified on the validation set. As detailed in Sec. 3.2, in order to ensure the validity of scaling, it is essential for the percentile value $p$ to fall within a specific range where the parameter $C(p)$ exhibits a sufficiently high value to meet the required condition. Our experimental observations align with this theoretical premise (Tab. 4). Specifically, we have empirically observed that, up to the $85^{th}$ percentile threshold, the AUROC values for both Near-OOD and Far-OOD scenarios consistently show an upward trend. However, a noticeable decline becomes apparent beyond this percentile threshold. This empirical finding corroborates our theoretical insight, indicating that the parameter $C(p)$ experiences a reduction in magnitude as $p$ approaches the $90^{th}$ percentile.

| $p$ | 65 | 70 | 75 | 80 | 85 | 90 | 95 |
|---|---|---|---|---|---|---|---|
| Near-OOD | 62.45 / 79.31 | 61.65 / 79.83 | 61.12 / 80.41 | 60.12 / 81.01 | **59.76 / 81.36** | 63.19 / 80.14 | 78.62 / 73.40 |
| Far-OOD | 24.08 / 94.43 | 22.21 / 95.02 | 20.20 / 95.61 | 18.26 / 96.17 | **16.53 / 96.53** | 18.58 / 96.20 | 32.42 / 93.28 |

Table 4: FPR@95 / AUROC results on ImageNet benchmarks under different $p$.

### 4.3 ISH FOR TRAINING-TIME MODEL ENHANCEMENT

We used the same dataset splits as the post-hoc experiments in Sec. 4.1. For training, we fine-tuned the torchvision pretrained model with ISH for 10 epochs with a cosine annealing learning rate schedule

initiated at 0.003 and a minimum of 0. We additionally observed that using a smaller weight decay value (5e-6) enhances OOD detection performance. The results are presented in Tab. 5. We compare ISH with other training-time model enhancement methods.

**Comparison with training-time methods.** The work LogitNorm (Wei et al., 2022) focuses on diagnosing the gradual narrowing of the gap between the logit magnitudes of ID and OOD distributions during later stages of training. Their proposed approach involves normalizing logits, and the scaling factor is applied within the logits space during the backward pass.

The key distinction between their LogitNorm method and our ISH approach lies in the purpose of scaling. LogitNorm scales logits primarily for confidence calibration, aiming to align the model's confidence with the reliability of its predictions. In contrast, ISH scales activations to prioritize weighted optimization, emphasizing the impact of high ID-ness data in the fine-tuning process.

**Comparison with data augmentation-based methods.** Zhang et al. (2023) indicate that data augmentation methods, while not originally designed for OOD detection improvement, can simultaneously enhance both ID and OOD detection accuracy. In comparison to AugMix and RegMixup, our ISH approach, while slightly inferior in ID accuracy, delivers superior OOD detection performance with significantly fewer computational resources. When compared to AugMix, ISH achieves substantial improvements, enhancing AUROC by 0.46 and 0.80 for Near-OOD and Far-OOD, respectively, with just 0.1x the extended training epochs. Notably, ISH sets the highest AUROC records, reaching 84.01% on Near-OOD scores and 96.79% on Far-OOD scores among all methods on the OpenOODv1.5 benchmark.

| Model | Training | Epochs Ori.+Ext. | Postprocessor | Near-OOD | | Far-OOD | | ID ACC |
|---|---|---|---|---|---|---|---|---|
| | | | | FPR@95 ↓ | AUROC ↑ | FPR@95 ↓ | AUROC ↑ | ↑ |
| ResNet-50 | LogitNorm (Wei et al., 2022) | 90+30 | MSP | 68.56 | 74.62 | 31.33 | 91.54 | 76.45 |
| | CIDER (Ming et al., 2023) | 90+30 | KNN | 71.69 | 68.97 | 28.69 | 92.18 | - |
| | TorchVision Model | 90 | SCALE | 59.76 | 81.36 | 16.53 | 96.53 | 76.13 |
| | TorchVision Model Extended | 90+10 | SCALE | 59.25 | 82.67 | 18.48 | 96.24 | 76.84 |
| | AugMix (Hendrycks et al., 2020) | 180 | SCALE | 60.58 | 83.55 | 21.01 | 95.99 | **77.64** |
| | RegMixup (Pinto et al., 2022) | 90+30 | SCALE | 63.55 | 80.85 | 19.87 | 95.94 | 76.88 |
| | **ISH (Ours)** | 90+**10** | SCALE | **55.73** | **84.01** | **15.62** | **96.79** | 76.74 |

Table 5: **Comparisons between our training-time ISH and state-of-the-art methods on ImageNet-1K.** Our ISH method achieves the highest scores for both Near-OOD and Far-OOD with the least training epochs. "Ori." denotes the original training epochs for the pretrained network, while "Ext." denotes the extended training epochs in our training scheme.

## 5 CONCLUSION

In this paper, we have conducted an in-depth investigation into the efficacy of scaling techniques in enhancing out-of-distribution (OOD) detection. Our study is grounded in the analysis of activation distribution disparities between in-distribution (ID) and OOD data. To this end, we introduce SCALE, a post-hoc model enhancement method that achieves state-of-the-art OOD detection accuracy when integrated with energy scores, without compromising ID accuracy. Furthermore, we extend the application of scaling to the training phase, introducing ISH, a training-time enhancement method that significantly bolsters OOD detection accuracy.

## 6 ACKNOWLEDGEMENTS

This research is supported by the National Research Foundation, Singapore under its NRF Fellowship for AI (NRF-NRFFAI1-2019-0001). Any opinions, findings and conclusions or recommendations expressed in this material are those of the author(s) and do not reflect the views of National Research Foundation, Singapore.

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

## A  DETAILS OF PROOF

**Proposition 3.1.** *Assume that ID activations $a_j^{ID} \sim \mathcal{N}^R(\mu^{ID}, \sigma^{ID})$ and OOD activations $a_j^{OOD} \sim \mathcal{N}^R(\mu^{OOD}, \sigma^{OOD})$ where $\mathcal{N}^R$ denotes a rectified Gaussian distribution. If $\mu^{ID}/\sigma^{ID} > \mu^{OOD}/\sigma^{OOD}$, then there is a range of percentiles $p$ for which a factor $C(p) = \frac{\varphi(\sqrt{2}\,\mathrm{erf}^{-1}(2p-1))}{1-\Phi(\sqrt{2}\,\mathrm{erf}^{-1}(2p-1))}$ is large enough such that $Q_p^{ID}/Q^{ID} < Q_p^{OOD}/Q^{OOD}$.*

*Proof.* The proof schema is to derive equivalent conditions. Under the assumption that data in the latent space follows an independent and identically distributed (IID) Gaussian distribution prior to the ReLU activation (Sun et al. (2021)), we can derive that each coefficient $a_j^{ID} \sim \mathcal{N}^R(\mu^{ID}, \sigma^{ID})$ and OOD activations $a_j^{OOD} \sim \mathcal{N}^R(\mu^{OOD}, \sigma^{OOD})$ where $\mathcal{N}^R$ denotes a rectified Gaussian distribution. Moreover if we denote high activation $h_j^{ID} = a_j^{ID}$ if $a_j > P_p(a)$ and zeros elsewhere. Then we have $h_j^{ID} \sim \mathcal{N}^T(\mu^{ID}, \sigma^{ID})$ and identically $h_j^{OOD} \sim \mathcal{N}^T(\mu^{OOD}, \sigma^{OOD})$, where $\mathcal{N}^T$ denotes a truncated Gaussian distribution. Then, we can calculate the expectations as follows:

$$\mathbb{E}[a_j] = \mu \left[ 1 - \Phi(-\frac{\mu}{\sigma}) \right] + \varphi(-\frac{\mu}{\sigma})\sigma \tag{9}$$

$$\mathbb{E}[h_j] = \mu + \frac{\varphi(m)}{1 - \Phi(m)}\sigma, \ m = \frac{s - \mu}{\sigma} \tag{10}$$

Here, $\varphi(\cdot)$ is the probability density function of the standard normal distribution, and $\Phi(\cdot)$ is its cumulative distribution function.

$Q_p/Q = \frac{\sum_j h_j}{\sum_j a_j} = \frac{\mathbb{E}[h_j](1-p)D}{\mathbb{E}[a_j]D}$. Let us consider the notation $\beta = (1-p)Q/Q_p = \frac{\mathbb{E}[a_j]}{\mathbb{E}[h_j]}$. $Q_p^{ID}/Q^{ID} < Q_p^{OOD}/Q^{OOD} \iff \beta^{ID} > \beta^{OOD}$. So we focus on:

$$\beta = \frac{\mu \left[ 1 - \Phi(-\frac{\mu}{\sigma}) \right] + \varphi(-\frac{\mu}{\sigma})\sigma}{\mu + \frac{\varphi(m)}{1 - \Phi(m)}\sigma} = \frac{1 - \Phi(-\frac{\mu}{\sigma})}{1 + \frac{\varphi(m)}{1 - \Phi(m)}\frac{\sigma}{\mu}} + \frac{\varphi(-\frac{\mu}{\sigma})\sigma}{\mu + \frac{\varphi(m)}{1 - \Phi(m)}\sigma} \tag{11}$$

Let's introduce some notations for ease of analysis:

- $\gamma = \frac{\mu}{\sigma}$

- $A = \Phi(-\gamma)$

- $B = \varphi(-\gamma)$

- $C = \frac{\varphi(m)}{1 - \Phi(m)} = \frac{\varphi(\frac{s-\mu}{\sigma})}{1 - \Phi(\frac{s-\mu}{\sigma})},$

With these definitions, we can express $\beta$ as:

$$\beta = \frac{1 - A}{1 + C\gamma^{-1}} + \frac{B\sigma}{\mu + C\sigma} \tag{12}$$

We consider that $\gamma^{ID} \geq \gamma^{OOD}$ Hence, we also have:

- $A^{ID} \leq A^{OOD}$

- $B^{ID} \leq B^{OOD}$

By definition we have that $s^{ID}(p) = \mu^{ID} + \sigma^{ID}\sqrt{2}\operatorname{erf}^{-1}(2p-1)$ and $s^{OOD}(p) = \mu^{OOD} + \sigma^{OOD}\sqrt{2}\operatorname{erf}^{-1}(2p-1)$ where $p$ is the proportion of data that we want to keep. So we have:

$$C^{ID}(p) = \frac{\varphi(m^{ID})}{1 - \Phi(m^{ID})} = \frac{\varphi(\frac{s^{ID}-\mu^{ID}}{\sigma^{ID}})}{1 - \Phi(\frac{s^{ID}-\mu^{ID}}{\sigma^{ID}})} = \frac{\varphi(\sqrt{2}\operatorname{erf}^{-1}(2p-1))}{1 - \Phi(\sqrt{2}\operatorname{erf}^{-1}(2p-1))} \tag{13}$$

Moreover, we can prove that $C^{OOD}(p) = \frac{\varphi(m^{OOD})}{1-\Phi(m^{OOD})} = \frac{\varphi(\frac{s^{OOD}-\mu^{OOD}}{\sigma^{OOD}})}{1-\Phi(\frac{s^{OOD}-\mu^{OOD}}{\sigma^{OOD}})} = \frac{\varphi(\sqrt{2}\operatorname{erf}^{-1}(2p-1))}{1-\Phi(\sqrt{2}\operatorname{erf}^{-1}(2p-1))} = C^{ID}(p)$.

Now, if we consider the approximation:

$$\mathbb{E}[a_j] \simeq \mu \left[ 1 - \Phi(-\frac{\mu}{\sigma}) \right] \tag{14}$$

We assume that $\varphi\left(-\frac{\mu}{\sigma}\right)\sigma \approx 0$ since the sigma term is very small, and the second term is below one. With this approximation, we have:

$$\beta = \frac{\gamma(1-A)}{\gamma+C} \tag{15}$$

We want to compare $\beta$ for in-distribution (ID) denoted $\beta^{ID}$ and out-of-distribution (OOD) data denoted $\beta^{OOD}$. Moreover, we have:

$$\beta^{ID} \geq \beta^{OOD} \iff \frac{1-A^{ID}}{1+C\gamma^{ID-1}} \geq \frac{1-A^{OOD}}{1+C\gamma^{OOD-1}} \iff \frac{1-A^{ID}}{1-A^{OOD}} \geq \frac{1+C\gamma^{ID-1}}{1+C\gamma^{OOD-1}} \tag{16}$$

We can use the approximation: $\frac{1}{1+C\gamma^{OOD-1}} \simeq 1-C\gamma^{OOD-1}$ by applying a first-order Taylor expansion. Then we have:

$$\frac{1-A^{ID}}{1-A^{OOD}} \geq (1+C\gamma^{ID-1})(1-C\gamma^{OOD-1}) \tag{17}$$

$$\geq 1 + C(\gamma^{ID-1} - \gamma^{OOD-1}) - C^2(\gamma^{ID-1}\gamma^{OOD-1}) \tag{18}$$

Note that by definition $C$ should be positive. The given inequality can be expressed as:

$$\frac{1-A^{ID}}{1-A^{OOD}} - 1 - C(\gamma^{ID-1} - \gamma^{OOD-1}) + C^2(\gamma^{ID-1}\gamma^{OOD-1}) \geq 0 \tag{19}$$

We can rewrite it as:

$$\partial_1 C^2 + \partial_2 C + \partial_3 \geq 0 \tag{20}$$

Here we have the following notations: $\partial_1 = (\gamma^{ID-1}\gamma^{OOD-1})$ and $\partial_2 = -(\gamma^{ID-1} - \gamma^{OOD-1})$ and $\partial_3 = \frac{1-A^{ID}}{1-A^{OOD}} - 1$. Let us define $\Delta = \partial_2^2 - 4\partial_1\partial_3$ Then we have:

$$\Delta = (\gamma^{ID-1} - \gamma^{OOD-1})^2 - 4(\gamma^{ID-1}\gamma^{OOD-1})\left(\frac{1-A^{ID}}{1-A^{OOD}} - 1\right) \tag{21}$$

$$= \gamma^{ID-2} + \gamma^{OOD-2} - 2(\gamma^{ID-1}\gamma^{OOD-1})\left(2\frac{1-A^{ID}}{1-A^{OOD}} - 1\right) \tag{22}$$

$$= \left(\gamma^{ID-1} + \gamma^{OOD-1}\right)^2 - 4(\gamma^{ID-1}\gamma^{OOD-1})\left(\frac{1-A^{ID}}{1-A^{OOD}}\right) \tag{23}$$

Since $\partial_1 > 0$, there are two possible cases:

- if $\Delta \leq 0$ then $C(p) \in \mathbb{R}^+$

- if $\Delta > 0$ then $C(p) \in \left[\max\left(\frac{(\gamma^{ID-1} - \gamma^{OOD-1})+\sqrt{\Delta}}{2(\gamma^{ID-1}\gamma^{OOD-1})}, 0^+\right), +\infty\right)$. Note that another side $(\gamma^{ID-1} - \gamma^{OOD-1}) \leq 0$ so $(\gamma^{ID-1} - \gamma^{OOD-1}) - \sqrt{\Delta} \leq 0$. So we do not consider this.

In summary, there is a valid range of pruning $p$ value satisfying the valid range of $C(p)$ so that the statistics $Q_p/Q$ of the ID distribution is smaller than that of the OOD distributions. $p$ with a larger $C(p)$ is more applicable to any case. $\square$

## B  FULL EXPERIMENTS

In this section, we provide full results for SCALE post-hoc model enhancement. Table 6 shows full results on ImageNet and Tab. 8 and 9 show full results on CIFAR10 and CIFAR100. We also provide ImageNet results following the dataset setting of ReAct and ASH in Tab. 7 for more comparison.

| Method | Near-OOD | | | Far-OOD | | | | ID ACC |
|---|---|---|---|---|---|---|---|---|
| | SSB-hard | NINCO | Average | iNaturalist | Textures | OpenImage-O | Average | |
| EBO | 76.54 / 72.08 | 60.59 / 79.70 | 68.56 / 75.89 | 31.33 / 90.63 | 45.77 / 88.7 | 38.08 / 89.06 | 38.40 / 89.47 | 76.18 |
| MSP | 74.49 / 72.09 | 56.84 / 79.95 | 65.67 / 76.02 | 43.34 / 88.41 | 60.89 / 82.43 | 50.16 / 84.86 | 51.47 / 85.23 | 76.18 |
| MLS | 76.19 / 72.51 | 59.49 / 80.41 | 67.84 / 76.46 | 30.63 / 91.16 | 46.11 / 88.39 | 37.86 / 89.17 | 38.20 / 89.58 | 76.18 |
| GEN | 75.72 / 72.01 | 54.88 / 81.70 | 65.30 / 76.85 | 26.12 / 92.44 | 46.23 / 87.60 | 34.52 / 89.26 | 35.62 / 89.77 | 76.18 |
| RMDS | 77.88 / 71.77 | 52.20 / 82.22 | 65.04 / 76.99 | 33.67 / 87.24 | 48.80 / 86.08 | 40.27 / 85.84 | 40.91 / 86.38 | 76.18 |
| TempScale | 73.90 / 72.87 | 55.12 / 81.41 | 64.51 / 77.14 | 37.70 / 90.50 | 56.92 / 84.95 | 45.39 / 87.22 | 46.67 / 87.56 | 76.18 |
| ReAct | 77.57 / 73.02 | 55.92 / 81.73 | 66.75 / 77.38 | 16.73 / 96.34 | 29.63 / 92.79 | 32.58 / 91.87 | 26.31 / 93.67 | 75.58 |
| ASH-S | 70.80 / 74.72 | 53.26 / 84.54 | 62.03 / 79.63 | 11.02 / 97.72 | **10.90 / 97.87** | 28.60 / 93.82 | 16.86 / 96.47 | 75.51 |
| **SCALE (Ours)** | **67.72 / 77.35** | **51.80 / 85.37** | **59.76 / 81.36** | **9.51 / 98.02** | 11.90 / 97.63 | **28.18 / 93.95** | **16.53 / 96.53** | **76.18** |

Table 6: FPR@95 / AUROC for ResNet-50 on ImageNet on OpenOOD v1.5 benchmark.

| Method | iNaturalist | | SUN | | Places365 | | Textures | | Average | | ID ACC |
|---|---|---|---|---|---|---|---|---|---|---|---|
| | FPR@95 ↓ | AUROC ↑ | FPR@95 ↓ | AUROC ↑ | FPR@95 ↓ | AUROC ↑ | FPR@95 ↓ | AUROC ↑ | FPR@95 ↓ | AUROC ↑ | ↑ |
| MSP | 54.99 | 87.74 | 70.83 | 80.86 | 73.99 | 79.76 | 68.00 | 79.61 | 66.95 | 81.99 | 76.12 |
| EBO | 55.72 | 89.95 | 59.26 | 85.89 | 64.92 | 82.86 | 53.72 | 85.99 | 58.41 | 86.17 | 76.12 |
| ReAct | 20.38 | 96.22 | 24.20 | 94.20 | **33.85** | 91.58 | 47.30 | 89.80 | 31.43 | 92.95 | - |
| DICE | 25.63 | 94.49 | 35.15 | 90.83 | 46.49 | 87.48 | 31.72 | 90.30 | 34.75 | 90.77 | - |
| DICE + ReAct | 18.64 | 96.24 | 25.45 | 93.94 | 36.86 | 90.67 | 28.07 | 92.74 | 27.25 | 93.40 | - |
| ASH-S | 11.49 | 97.87 | 27.98 | 94.02 | 39.78 | 90.98 | **11.93** | **97.60** | 22.80 | **95.12** | 74.98 |
| **SCALE (Ours)** | **9.50** | **98.17** | **23.27** | **95.02** | 34.51 | **92.26** | 12.93 | 97.37 | **20.05** | **95.71** | **76.12** |

Table 7: Results for ResNet-50 on ImageNet following same testing splits as Sun et al. (2021).

| Method | SVHN | | iSUN | | Textures | | Places365 | | Average | | ID ACC |
|---|---|---|---|---|---|---|---|---|---|---|---|
| | FPR@95 ↓ | AUROC ↑ | FPR@95 ↓ | AUROC ↑ | FPR@95 ↓ | AUROC ↑ | FPR@95 ↓ | AUROC ↑ | FPR@95 ↓ | AUROC ↑ | ↑ |
| MSP | 47.24 | 93.48 | 42.31 | 94.52 | 64.15 | 88.15 | 63.02 | 88.57 | 54.18 | 91.18 | 94.53 |
| EBO | 40.61 | 93.99 | 10.07 | 98.07 | 56.12 | 86.43 | 39.40 | 91.64 | 36.55 | 92.53 | 94.53 |
| ReAct | 41.64 | 93.87 | 12.72 | 97.72 | 43.58 | 92.47 | 43.31 | 91.03 | 35.31 | 93.77 | - |
| DICE | $25.99^{\pm5.10}$ | $95.90^{\pm1.08}$ | $4.36^{\pm0.71}$ | $99.14^{\pm0.15}$ | $41.90^{\pm4.41}$ | $88.18^{\pm1.80}$ | $48.59^{\pm1.53}$ | $89.13^{\pm0.31}$ | $30.21^{\pm2.94}$ | $93.09^{\pm0.84}$ | - |
| ASH-S | 6.51 | 98.65 | 5.17 | 98.90 | 24.34 | **95.09** | 48.45 | 88.34 | 21.12 | 95.25 | 94.02 |
| **SCALE (Ours)** | **5.80** | **98.72** | **3.43** | **99.21** | 23.42 | 94.97 | **38.69** | **91.74** | **17.84** | **96.16** | 94.53 |

Table 8: Detailed results for CIFAR-10.

| Method | SVHN | | iSUN | | Textures | | Places365 | | Average | | ID ACC |
|---|---|---|---|---|---|---|---|---|---|---|---|
| | FPR@95 ↓ | AUROC ↑ | FPR@95 ↓ | AUROC ↑ | FPR@95 ↓ | AUROC ↑ | FPR@95 ↓ | AUROC ↑ | FPR@95 ↓ | AUROC ↑ | ↑ |
| MSP | 81.70 | 75.40 | 85.99 | 70.17 | 84.79 | 71.48 | 82.55 | 74.31 | 83.76 | 72.84 | 75.04 |
| EBO | 87.46 | 81.85 | 74.54 | 78.95 | 84.15 | 71.03 | 79.20 | 77.72 | 81.34 | 77.39 | 75.04 |
| ReAct | 83.81 | 81.41 | 65.27 | 86.55 | 77.78 | 78.95 | 82.65 | 74.04 | 77.38 | 80.24 | - |
| DICE | $54.65^{\pm4.94}$ | $88.84^{\pm0.39}$ | $48.72^{\pm1.55}$ | $90.08^{\pm1.36}$ | $65.04^{\pm0.66}$ | $76.42^{\pm0.35}$ | $79.58^{\pm2.34}$ | $77.26^{\pm1.08}$ | $62.00^{\pm2.37}$ | $83.15^{\pm0.80}$ | - |
| ASH-S | 25.02 | 95.76 | 46.67 | 91.30 | **34.02** | **92.35** | 85.86 | 71.62 | 47.89 | 87.76 | 71.65 |
| **SCALE (Ours)** | **22.05** | **96.29** | **42.14** | **92.47** | 34.20 | 92.34 | **85.04** | **72.66** | **45.86** | **88.44** | 75.04 |

Table 9: Detailed results for CIFAR-100.

