# OpenReview forum: "Scaling for Training Time and Post-hoc Out-of-distribution Detection Enhancement"
_ICLR.cc/2024/Conference — ICLR 2024 poster_

### Official Review · Reviewer_5qNg · 2023-10-25

**Soundness:** 2 fair
**Presentation:** 3 good
**Contribution:** 2 fair
**Rating:** 5
**Confidence:** 4

**Summary:**

This paper investigates modern deep learning systems' ability to identify out-of-distribution (OOD) samples. It critically analyzes the extremely simple activation shaping (ASH) method, finding that activation pruning hinders, while activation scaling improves OOD detection. The authors propose two methods: SCALE, a post-training enhancement that boosts OOD detection without affecting in-distribution accuracy, and Intermediate Tensor SHaping (ISH), a training-time method for enhancing OOD detection. These methods show significant performance improvements on the OpenOOD v1.5 ImageNet-1K benchmark.

**Strengths:**

1. The analysis of ASH looks reasonable.
2. SCLAE and ISH are easy to reproduce and use.
3. It is commendable that similar ideas can be applied simultaneously to post-hoc and training time OOD detection.

**Weaknesses:**

1. Incorrect Motivation: The author seems to have not conducted sufficient research in the field of OOD detection. Generally, OOD detection requires the model to identify OOD data without affecting the accuracy of ID classification. Hence, there should not be a trade-off between ID classification accuracy and OOD detection performance. Post-hoc OOD detection refers to performing OOD detection through post-processing algorithms without altering the model parameters. Consequently, React, Dice, and ASH  utilize the original classifier for ID classification tasks and a modified classifier for OOD detection tasks, introducing a very minimal computational overhead. Therefore, I believe there is a problem with the motivation presented in the introduction section of the paper.
2. Limited Novelty: SCALE and ISH are incremental works of ASH. Thus, I attribute the commendable performance of ISH and SCALE mainly to the superior performance of ASH. The performance improvement over ASH is quite minimal.
3. Unsubstantiated Claim on React: The author claims that React “hinders the OOD detection process.” However, this analysis is solely theoretical and lacks experimental validation. I think the results from SCALE+React or ISH+React could potentially validate this point.
4. Minimal Performance Gain: As observed from Table 5, the performance improvement brought by combining ISH and SCALE compared to using SCALE alone is extremely minimal. I am curious to know if combining ISH with other OOD detectors (e.g., ISH+MSP/ODIN/Energy/React/ASH) could yield any performance improvement.
5. Lack of Ablation Study: The paper is missing an ablation study related to the placement of SCALE, specifically, how its performance varies when placed after different stages of ResNet50.

**Questions:**

see Weakness

---

> ### Author Response · Authors · 2023-11-22
>
> We express our appreciation to reviewer 5qNg for your insightful review and comprehensive suggestions.
> - **Incorrect Motivation: ID-OOD accuracy trade-off:**
>
> Thanks for highlighting this confusion. Indeed the ID accuracy drop can be avoided through a separate classifiers with minimal computational cost. We will emphasize this point more in the introduction. We note that ID-OOD accuracies trade-offs are also reported in ReAct Table 2 and DICE Table 5, and further explored in ASH where they argue that an ideal OOD detection pipeline "should not deteriorate ID task performance, nor should it require a cumbersome parallel setup that handles the ID task and OOD detection separately". We will clarify this point.
> We note that our motivation is not only achieving a better tradeoff between OOD detection and ID classification but also achieving higher OOD detection performance than others. We do this in a principled way through theoretical analysis, giving explanations and insights into the working mechanisms of existing works before making our proposal.
>
> - **Limited Novelty of SCALE and ISH:**
>
> Thank you for your comments. The main contribution is providing a theoretical understanding of the current state-of-the-art method ASH. Our development of SCALE is directly born out of this analysis and verifies that it is only the scaling operation which benefits OOD detection (as pointed out by reviewer AeGx).
> Furthermore, we propose to leverage scores as ID-ness in extra training, which further improves OOD detection without the need for extra OOD validation data.
> For experimental results. Compared to ASH-S, we achieved 1.73+ on near-OOD, +0.06 on far-OOD, and +0.67 on ID accuracy for SCALE, and achieved 4.83+ on near-OOD, 0.32+ on far-OOD, and 0.56+ on ID accuracy for ISH+SCALE. We note that current methods are already good for easier far-OOD detection so the improvement room is smaller.
>
> - **Unsubstantiated Claim on ReAct:**
>
> Thanks for your suggestions. Our statement in the introduction is that ”the lower-part pruning approach, in contrast to ReAct, hinders the OOD detection process”. ReAct is a higher-part pruning. In our original paper, we provide an analysis that ASH pruning impedes the OOD process because lower-part pruning brings ID and OOD data closer in the logits space. For ReAct, higher-part pruning enlarges the distance between ID and OOD and this provides an explanation for its effectiveness.
>
>
> - **Minimal Performance Gain and combination of ISH with other OOD detector:**
>
> Thanks for your suggestions. For the experiments results, ISH + SCALE improves an additional notable 1.34 on SCALE, which has 21% improvement than AugMix with less training efforts. We provide results for ISH + MSP/ODIN/Energy/ReAct/ASH in the following table and show that ISH increases all results.
>
> |            | Near-OOD AUROC | Far-OOD AUROC |
> |------------|----------------|---------------|
> | MSP        |    76.03       |     85.23     |
> | MSP+ISH    |    **76.87**       |     **85.86**     |
> | ODIN       |     74.75      |     89.47     |
> | ODIN+ISH   |     **76.51**          |    **90.18**        |
> | Energy     |    75.89       |     89.47     |
> | Energy+ISH |     **77.62**      |     **90.06**     |
> | ReAct      |     77.38      |     93.67     |
> | ReAct+ISH  |     **78.99**      |     **93.96**     |
> | ASH-S        |     79.63      |     96.47     |
> | ASH-S+ISH    |     **82.26**      |     **96.61**     |
>
> - **Ablation Study for the placement of SCALE :**
>
> We appreciate your valuable suggestions and provide the ablation study on the following table:
>
>
> |             | Near-OOD AUROC | Far-OOD AUROC |
> |-------------|---------------|---------------|
> | Block1      |  48.62  |  64.09  |
> | Block2      |  62.79  |    62.06 |
> | Block3      | 70.08   |  62.27  |
> | Block4      | 78.86   |  90.65  |
> | Penultimate | **81.36** | **96.53**   |
>
> As the stage becomes shallower, the performance decreases. This is because the early stages tend to learn low-level features that are common to both the ID and OOD datasets.

---

### Official Review · Reviewer_AeGx · 2023-10-29

**Soundness:** 4 excellent
**Presentation:** 4 excellent
**Contribution:** 4 excellent
**Rating:** 8
**Confidence:** 5

**Summary:**

This work studies the problem of OOD detection. It starts by analyzing a recent SOTA method, ASH, where the analysis successfully disentangles the effect of ASH's two building components, namely pruning and scaling. Surprisingly, it is found that while scaling benefits the separation of ID and OOD score, pruning actually adversely hurts it. Based on this analysis, a new post-hoc method named SCALE is proposed. Furthermore, a training-time regularization that echoes the working mechanism of SCALE is also proposed (ISH). Both SCALE and ISH achieves improvements over closely-related and competitive baselines on multiple benchmarks.

**Strengths:**

1. The written and presentation quality of this manuscript is good. Easy to follow, and the logic is smooth and sound.
2. The analysis of ASH and the insights originate from the analysis are valuable for two reasons. First, ASH indeed leads to superior performance (a recent SOTA), and thus understanding why it works is of course important. Second, the finding that scaling actually benefits OOD detection the most is surprising. In a bigger picture, I think this finding contradicts the common belief that pruning / rectifying is the (only) right direction (this can be evidenced by that many methods, e.g., ReAct, DICE, ASH, adopt such general design idea).
3. The analysis is backed up by both theoretical investigation (Proposition 3.1) and empirical evidences (Table 1 and Figure 2 - 4).
4. The developed method is motivated by the analysis, making it sound and valid.
5. The empirical improvement on OpenOOD ImageNet-1K benchmark is convincing. In the first place, I also like it that the authors consider a unified benchmark for evaluation, which many works failed to do. This will definitely encourage follow-up works to continue making fair and straight comparison, which would benefits the whole OOD detection community.

**Weaknesses:**

I have two minor comments.

1. For CIFAR experiments, I strongly encourage the authors to refrain from using LSUN-Crop and LSUN-Resize as OOD datasets, if they are directly taken from the ODIN paper. LSUN-Resize has been shown to exhibit obvious resizing artifacts [1], which make the detection trivially easy. LSUN-Crop is 32x32 crops from images with larger resolution, and arguably the resulting samples will have unnaturally different distribution compared to CIFAR's 32x32 natural images, which again makes the evaluation less meaningful.

2. The "Comparison with OOD traning methods." subtitle in Sec. 4.3 has a typo ("traning" -> "training"), and should be placed in the same line as the leading sentence of the next paragraph. There also should be a blank space in "LogitNorm(Wei et al., 2022)". I would suggest a proof-reading to further improve the quality.

[1] CSI: Novelty Detection via Contrastive Learning on Distributionally Shifted Instances

**Questions:**

I was wondering if the analysis provided in this work can also help explain ReAct's effect. Some elaboration or discussion on this can help provide even a more complete picture.

---

> ### Author Response · Authors · 2023-11-22
>
> We extend our gratitude to RAx2 for the insightful review and suggestions.
>
> - **LSUN-Crop and LSUN-Resize results**:
> Thanks for pointing this out. In our revision, we will remove LSUN-Crop and LSUN-Resize from the main text and place the original table in the Supplementary for consistent comparison to ODIN; and provide discussion to give awareness to the reader.
>
>
>
> - **Typos and mistakes**: Thank you for pointing out these. We will fix them in the revision and proofread  them in the future carefully.
>
>
>
> - **Explanation of ReAct**:
> Our analysis is relevant to ReAct, a higher-part pruning approach. In our original paper, we provide an analysis that ASH pruning impedes the OOD process because lower-part pruning brings ID and OOD data closer in the logits space. For ReAct, higher-part pruning enlarges the distance between ID and OOD and this provides an explanation for its effectiveness.

---

> > ### Comment · Reviewer_AeGx · 2023-11-22
> > **Thanks for the rebuttal**
> >
> > I have read the authors' response. I maintain my score of 8 and definitely recommend for acceptance.

---

> > > ### Author Response · Authors · 2023-11-23
> > >
> > > We express our gratitude for your acknowledgement and constructive feedback. Thank you for your time and effort in reviewing our work.

---

### Official Review · Reviewer_DQbC · 2023-11-02

**Soundness:** 3 good
**Presentation:** 4 excellent
**Contribution:** 4 excellent
**Rating:** 6
**Confidence:** 3

**Summary:**

The authors present Intermediate Tensor SHaping (ISH) method for efficient OOD detection.

**Strengths:**

The method presented in this work is interesting and innovative. I appreciate the idea and encourage authors to further the work along this domain.

**Weaknesses:**

The method is tested only in Imagenet. The authors should use it for OOD detection in other kinds of datasets to ensure that the method is applicable in various domains.

**Questions:**

How does the method generalize to other types of data e.g. tabular, time-series etc.?

---

> ### Author Response · Authors · 2023-11-22
>
> We thank reviewer DQbC for your insightful review.
> - **The method is tested only in Imagenet:**
> Table 3 shows experiments on CIFAR-10 and CIFAR-100.
> - **Other Data Domains:**
> Our method, like prevailing ad-hoc methods for OOD (EBO, ReAct, ASH) are intended for already-trained networks by design. The readily available off-the-shelf image backbones is why we and others work primarily in the image domain, extending it to other domains like time series and tabular data which do not have analogous backbones will likely need fundamental changes in the assumptions and approach beyond our current scope; we leave this for future work.
> We strongly support the point that OOD for other domains is a worthwhile focus and we leave it for future research.
>
> EBO: Liu, Weitang, Xiaoyun Wang, John Owens, and Yixuan Li. "Energy-based out-of-distribution detection." _Advances in neural information processing systems_ 33 (2020): 21464-21475.
>
> ReAct: Sun, Yiyou, Chuan Guo, and Yixuan Li. "React: Out-of-distribution detection with rectified activations." _Advances in Neural Information Processing Systems_ 34 (2021): 144-157.
>
> ASH: Djurisic, Andrija, Nebojsa Bozanic, Arjun Ashok, and Rosanne Liu. "Extremely simple activation shaping for out-of-distribution detection." _arXiv preprint arXiv:2209.09858_ (2022).

---

### Official Review · Reviewer_RAx2 · 2023-11-03

**Soundness:** 3 good
**Presentation:** 3 good
**Contribution:** 3 good
**Rating:** 6
**Confidence:** 3

**Summary:**

The authors first analyze the OOD detection method ASH and then propose a post-hoc network enhancement method for OOD detection, namely SCALE. Besides, they propose a Intermediate Tensor Shaping method for training time OOD detection enhancement.

**Strengths:**

1.	The proposed method is simple and easy to implement.
2.	The results of the proposed method are better than the SOTA method ASH.

**Weaknesses:**

1.	Could the authors provide an algorithm to demonstrate the overall pipeline of the proposed method? I am confused about when we should invoke the method in Section 3.4.
2.	Recently, Transformer-based models become more and more popular. Could the proposed method be applied to these models?

**Questions:**

I am not familiar with this OOD detection field. However I think the proposed method is simple and effective based on the current manuscript.

---

> ### Author Response · Authors · 2023-11-22
>
> We extend our gratitude to RAx2 for the insightful review.
> - **Algorithm for ISH pipeline**:
> Thanks for your suggestion. We provide pseudo-code of the algorithm here and will add it to Sec. 4.3 in the revision.
> ```
> function ISH(pretrained_model M, data D, learning_rate L, epochs I)
> 	for input, label in D:
> 		# forward pass get predicted results and intermediate activations.
> 		activation_1, ...,  activations_p, predicted = forward_pass(M, input)
> 		gradient_z = loss(predicted-input)
>
> 		# get scaling s for the activations of for penultimate layer:
> 		s = SCALE(activations_p) # as defined in equation (7)
>
> 		# for penultimate layer:
> 		# scale activations_p by s for gradient computation
> 		gradient_weights = dot_product(s  * activations_p^T, gradient_z)
> 		gradient_biases = gradient_z
>
> 		update_parameters(gradient_weights, gradient_biases, learning_rate)
>
> 		update other parameters
>
> 		return weights, biases
> ```
> - **Applicability to Transformers**:
> Our current assumption and analysis is made in light of the ReLU because ResNet 50 is still the prevailing architecture of choice for other works in OOD [EBO, ReAct, ASH]. However, similar ideas could be applied to transformers, and we are looking into this for future work.
>
> **References**
>
> EBO: Liu, Weitang, Xiaoyun Wang, John Owens, and Yixuan Li. "Energy-based out-of-distribution detection." _Advances in neural information processing systems_ 33 (2020): 21464-21475.
>
> ReAct: Sun, Yiyou, Chuan Guo, and Yixuan Li. "React: Out-of-distribution detection with rectified activations." _Advances in Neural Information Processing Systems_ 34 (2021): 144-157.
>
> ASH: Djurisic, Andrija, Nebojsa Bozanic, Arjun Ashok, and Rosanne Liu. "Extremely simple activation shaping for out-of-distribution detection." _arXiv preprint arXiv:2209.09858_ (2022).

---

> > ### Comment · Reviewer_RAx2 · 2023-11-23
> > **Responses to Authors**
> >
> > I have no more questions and keep my scoring.

---

### Author Response · Authors · 2023-11-22

Dear meta reviewer and reviewers,

We appreciate your time and effort in reviewing our submission and providing detailed comments and valuable suggestions.


We are glad to find that reviewers like our work, specifically:
- Our method is simple, widely applicable, and effective in achieving SOTAs (RAx2, 5qNg).
- The findings of rigorous analysis (5qNg) and contributions are interesting, innovative, important, and promising to the community (DQbC, AeGx).
- The writing and presentation are good (RAx2, DQbC, AeGx, 5qNg); the motivation, logic, theory, and experiments on a unified benchmark are sound (AeGx).

 In response to the feedback, we will implement the following changes in the revision:
 - **[RAx2 Weak 1: Algorithm Illustration]** An algorithm of the overall pipeline of ISH + SCALE in Sec. 3.4, where we provided in the response for RAx2  .
 - **[AeGx Weak 1: Remove Redundant Results]** Remove LSUN-Crop and LSUN-Resize from the main text due to their obvious resizing artifacts.
 - **[5qNg Weak 4&5: Additional Results]** Additional ablation studies including layer placement and the combination for ISH with other OOD detectors.

We address remaining concerns in the reviewers-specific response. We add several new experiments as requested by reviewer 5qNg. All of these results further bolster support to the robustness and generalizability of our proposed method.



Thanks again and sincerely,

The authors

---

### Meta-Review · Area_Chair_9YTG · 2023-12-10

**Metareview:**

The paper starts by trying to understand the working principles behind ASH, an existing OOD detection algorithm. In the process of that, the authors discovered a better algorithm that emphasizes the significant role of scaling the activations. The process is an inspiring one to the community. The proposed method is simple, directly and widely applicable, and effective in achieving SoTA on various datasets and architectures.

The concerns about this paper is its potential lack of innovation, as it mostly leverages the framework of ASH. However we also recognize the importance of careful analyses of existing work, and promising discoveries as a result of such analyses.

**Justification For Why Not Higher Score:**

While performing a rigorous study, the paper's idea is set up as an incremental of ASH, a published work, and therefore its performance improvement primarily stems from the inherent superiority of ASH itself. In the same conference there are already many submissions providing newer methods outperforming ASH; while this paper does not necessarily need to compare to them, as they are concurrent submissions, in the long run the impact of ISH might be reduced because of that.

**Justification For Why Not Lower Score:**

The proposed method is simple, directly applicable, and effective in achieving SoTA.
The findings of rigorous analysis and contributions are interesting, innovative, important, and promising to the community.
The writing and presentation are good; the motivation, logic, theory, and experiments on a unified benchmark are sound.

---

### Decision · Program_Chairs · 2024-01-16

Accept (poster)